# Research Status and Prospect of Additive Manufactured Nickel-Titanium Shape Memory Alloys

**DOI:** 10.3390/ma14164496

**Published:** 2021-08-11

**Authors:** Shifeng Wen, Jie Gan, Fei Li, Yan Zhou, Chunze Yan, Yusheng Shi

**Affiliations:** 1State Key Laboratory of Material Processing and Die & Mold Technology, School of Materials Science and Engineering, Huazhong University of Science and Technology, Wuhan 430074, China; roya_wen@hust.edu.cn (S.W.); c_yan@hust.edu.cn (C.Y.); shiyusheng@hust.edu.cn (Y.S.); 2Faculty of Engineering, China University of Geosciences, Wuhan 430074, China; gj@cug.edu.cn; 3State Key Laboratory of Intelligent Manufacturing System Technology, Beijing Institute of Electronic System Engineering, Beijing 100854, China; lifei@casic.com.cn

**Keywords:** additive manufacturing (AM), nickel-titanium alloy, shape memory effect, superelastic, selective laser melting (SLM)

## Abstract

Nickel-titanium alloys have been widely used in biomedical, aerospace and other fields due to their shape memory effect, superelastic effect, as well as biocompatible and elasto-thermal properties. Additive manufacturing (AM) technology can form complex and fine structures, which greatly expands the application range of Ni-Ti alloy. In this study, the development trend of additive manufactured Ni-Ti alloy was analyzed. Subsequently, the most widely used selective laser melting (SLM) process for forming Ni-Ti alloy was summarized. Especially, the relationship between Ni-Ti alloy materials, SLM processing parameters, microstructure and properties of Ni-Ti alloy formed by SLM was revealed. The research status of Ni-Ti alloy formed by wire arc additive manufacturing (WAAM), electron beam melting (EBM), directional energy dedication (DED), selective laser sintering (SLS) and other AM processes was briefly described, and its mechanical properties were emphatically expounded. Finally, several suggestions concerning Ni-Ti alloy material preparation, structure design, forming technology and forming equipment in the future were put forward in order to accelerate the engineering application process of additive manufactured Ni-Ti alloy. This study provides a useful reference for scientific research and engineering application of additive manufactured Ni-Ti alloys.

## 1. Introduction

Nickel-titanium shape memory alloys (SMAs) have shape memory effect, superelastic effect, damping properties, biocompatibility and corrosion resistance [1,2], which have been widely used in aerospace, biomedical, automotive, construction and flexible electronics fields [3], as shown in Figure 1. However, the high ductility, rebound effect and work hardening properties of Ni-Ti alloy make it very challenging for machining to obtain high-quality and high-precision Ni-Ti alloy parts [4]. In addition, it is difficult to manufacture Ni-Ti alloy parts with complex structure by turning and milling, forging and casting, which greatly limits its application. Therefore, in order to realize the expansion of Ni-Ti alloy to complex shape structure and further improve its performance, it is urgent to seek a new forming technology for Ni-Ti alloy.

Additive manufacturing (AM) is a new manufacturing technology that reduces three-dimensional manufacturing to two-dimensional manufacturing by layering on top of each other. AM can be used to form parts of arbitrarily complex structures. In recent years, many scholars have carried out a large number of studies on additive manufactured Ni-Ti alloys [5,6,7]. Figure 2 shows the statistics of research papers published on additive manufactured Ni-Ti alloys in the Web of Science database in recent years (the statistical period from March 2010 to November 2020). As can be seen from Figure 2a, the number of papers published in this field is increasing year by year. With the improvement of AM equipment performance and the gradual optimization of AM forming process, relevant research projects have shown a significant growth trend in the last three years, especially since 2020. Figure 2b shows the ranking of the number of papers published in this field by different countries and regions. It shows that the United States started early in the research on additive manufactured Ni-Ti alloys and is also the main research front in this field, followed by China.

## 2. Ni-Ti Alloys Formed by AM Technologies

At present, the research on additive manufactured Ni-Ti alloys has begun to take shape. Figure 3 shows the number of papers published in the Web of Science database for the different AM technologies used for manufacturing Ni-Ti alloys. It is clear that selective laser melting (SLM) technology has become the most common manufacturing technology for Ni-Ti alloys. Therefore, in this paper, the forming process parameters, microstructure morphology, mechanical properties and functional properties of Ni-Ti alloys formed by SLM technology are mainly discussed. Then, the studies related to the formation of Ni-Ti alloys by laser net shaping (LENS) and other AM technologies are briefly introduced. Based on the above technologies, the current main application fields of Ni-Ti alloys are listed. The internal relationship between material, process and performance of additive manufactured Ni-Ti alloys is summarized, the possible reasons for the gap in engineering application of additive manufactured Ni-Ti alloys are analyzed, and the possible solutions are proposed. This paper will provide useful reference for the research of additive manufactured Ni-Ti alloys.

### 2.1. Ni-Ti Alloys Formed by SLM Technology

In terms of forming materials, Ni-Ti powders for SLM process are mainly obtained by two methods: one is to mix pure nickel and pure titanium powders by mechanical ball milling with a certain mass ratio; the other method is to prepare Ni-Ti alloy powders by electrochemical induction melting gas atomization (EIGA) with the Ni-Ti alloy ingots. Figure 4 shows the atomic ratio and particle size distribution of Ni-Ti alloy powders used in SLM technology. It can be seen that the nickel atomic ratio of Ni-Ti alloy powders used in SLM technology in literature ranges from 40 at.% to 54 at.% (yellow area in Figure 4), covering titanium-rich, iso-atomic ratio and nickel-rich Ni-Ti alloy. However, it is mainly composed of nickel-rich Ni-Ti alloys with equal atomic ratio, especially those with near equal atomic ratio. In addition, the blue area in Figure 4 indicates that the particle size of Ni-Ti alloy powder for SLM is widely distributed in the range of 15~80 μm, mainly in the range of 20~55 μm.

In terms of forming process, the overlap of melting lines and the layer accumulation involve complex thermal cycles during the SLM process, which have significant influence on the microstructure evolution and phase transformation behavior of Ni-Ti alloy, and directly affect the forming quality, mechanical properties and functional properties of Ni-Ti alloy. Therefore, considering the characteristics of Ni-Ti alloy materials and SLM process, reasonable adjustment and optimization of SLM processing parameters are necessary to obtain good properties of Ni-Ti alloy. Inappropriate SLM processing parameters will lead to many defects in the Ni-Ti alloy, such as cracks, holes and incomplete melting of powder, as shown in Figure 5. Zhao et al. [8] pointed out that, under the SLM process with high power and low scanning speed, some deep and narrow concave keyholes would be generated inside the alloy, and the instability at the tip of the keyholes would make the holes near them evolve into defects. If the laser power is too low and the scanning speed is too high, the Ni-Ti alloy powder cannot be completely fused, and the existence of unfused particles in the alloy will increase its porosity and deteriorate its properties. These cracks, voids and unmelted particles caused by inappropriate process parameters will result in a certain degree of porosity within the NiTi alloy specimen and further affect the metallurgical bonding between the layers and also increase the surface roughness of the as-printed samples. When NiTi alloy components are subjected to the alternating loads, stress concentration is prone to be generated at these internal or external defects, leading to the sprouting of fatigue cracks. Therefore, optimizing the forming process and investigating post-treatment methods (e.g., heat treatment, sandblasting and so on) to reduce internal defects and external surface roughness are particularly important for the engineering applications of NiTi alloys.

Compared with the traditional technology, the microstructure of Ni-Ti alloy formed by SLM process has significant characteristics, mainly in the aspects of grain morphology, texture and precipitation. The crystalline morphology of Ni-Ti alloy formed by SLM is closely related to laser scanning strategy and molten pool temperature gradient. As shown in Figure 6, when Z-shaped scanning strategy with 90° interlayer rotation angle was adopted, the crystal grains on the forming surface were observed to be regular rectangular under the optical microscope, showing elongated characteristics in the deposition direction (side) [9,10,11,12]. In addition, there are fine equiaxed grains on the surface of the sample [13], and columnar grains of nanometer and submicron scale exist on the molten pool boundary. Affected by the temperature gradient of the molten pool, it can be seen that the columnar grains are vertically distributed along the molten pool boundary, and there are fine equiaxed grains near the molten pool, as shown in Figure 6e,f.

The phase transition temperature of Ni-Ti alloy is closely related to the Ni/Ti atomic ratio [2]. Therefore, the ablation and evaporation degree of nickel element in the Ni-Ti alloy can be controlled by adjusting the SLM processing parameters, so as to realize the precise control of phase transition temperature. Mathew et al. [14] and Yang et al. [11] showed that the initial temperature of martensite transition (M_s_) decreased with the increase in laser scanning speed when the laser energy density remained constant. Wang et al. [15] studied the influence of SLM parameters on the phase transition temperature of Ni-Ti alloy, and the results showed that the peak martensitic transition temperature (M_P_) decreased with the increase in laser power. Moghaddam et al. [16] took the scanning spacing of SLM processing parameters as the dependent variable to investigate the relationship between it and the phase transition temperature of Ni-Ti alloy. The results showed that the phase transition temperature decreased with the increase in scanning spacing. Later, Wang et al. [17] systematically studied the influence of various SLM processing parameters on the independence of phase transition curves of Ni-Ti alloy, as shown in Figure 7, and the results showed that the phase transition temperature of Ni-Ti alloy decreased with the increase in laser scanning speed and scanning spacing, and increased with the increase in laser power and energy density. Many studies have shown that the phase transition temperature of Ni-Ti alloy can be controlled by changing the SLM processing parameters, but the quantitative evaluation of its ablation and evaporation degree with Ni element is still to be explored, so that Ni-Ti alloy can be used in different temperature environments more flexibly.

In practical application, Ni-Ti alloy as a structural component needs to bear several kinds of loads, and its mechanical properties are particularly important. Figure 8 summarizes the compression and tensile properties of Ni-Ti alloy formed by SLM, recently. The compression strength of Ni-Ti alloy is up to 3.2 GPa and the elongation is over 40%, which is comparable to the compression performance of as-cast Ni-Ti alloy [13]. In terms of tensile properties of Ni-Ti alloy formed by SLM, Lu et al. [18] formed a Ni-Ti alloy without defects, containing ultrafine cellular grains and nano-Ti_2_Ni under high laser energy density, with tensile strength up to 776 MPa. Xiong et al. [19] obtained a Ni-Ti alloy with an elongation of 15.6% by optimizing the interlayer scanning strategy. By strictly controlling the oxygen content in the forming chamber of SLM equipment, Wang et al. [17] further improved the elongation of Ni-Ti alloy to 23%. In general, as many scholars have explored and understood more deeply about the interaction between laser and NiTi alloy materials, by continuously improving the forming process, the strength and ductility of the as-printed NiTi alloys can further meet the practical needs of engineering applications. However, from the statistical data, there is a wide distribution of compressive and tensile properties of NiTi alloys formed by SLM technology. Specifically, its compressive properties are much higher than its tensile properties; this is attributed to the unique layered fabrication method of AM, which leads to a preferred orientation of grain growth within the alloy, resulting in significant anisotropy of its macroscopic properties. Therefore, in engineering practice, the SLMed NiTi alloy should be applied with appropriate loading direction and loading method based on the full investigation of its polycrystalline orientation distribution.

The functional properties of Ni-Ti alloy are mainly reflected in its excellent shape memory effect and superelasticity. For the shape memory effect, Saedi et al. [10] conducted a thermal cycle test on SLM and ingot Ni-Ti alloy under 50–300 MPa compression stress to compare their shape memory recovery performance. The results showed that the recoverable strain of the two kinds of Ni-Ti alloy samples first increased and then decreased with the increase in compression stress. The shape memory properties of SLM Ni-Ti alloy (recoverable strain 3.47%) are better than those of ingot Ni-Ti alloy (recoverable strain 3.26%) at 300 MPa. Lu et al. [18] obtained a Ni-Ti alloy with excellent tensile properties under a high laser energy density (>155 J/mm^3^), and its shape memory recovery rate reached 98.7% under the condition of 5.12% tensile predeformation. By optimizing the interlayer scanning strategy, Xiong et al. [19] formed a Ni-Ti alloy with an elongation of 15.6%, and its shape memory recovery rate could reach 98% when the predeformation was 2%, and then decreased with the increase in the predeformation. The Ni-Ti alloy components formed by this scanning strategy can produce a shape memory recovery rate of 99% at room temperature, as shown in Figure 9a. Furthermore, Ma et al. [27] adjusted the phase transition temperature of different parts of the Ni-Ti alloy by adjusting the SLM processing parameters, and realized the multi-stage shape memory recovery behavior of the Ni-Ti alloy, as shown in Figure 9b.

For the superelastic effect, Saedi et al. [10] studied the superelastic effect of SLM Ni-Ti alloy, SLM solid solution Ni-Ti alloy and ingot Ni-Ti alloy at different temperatures, and the results showed that the superelastic response of SLM and ingot Ni-Ti alloy was roughly the same, while the SLM solid solution Ni-Ti alloy presented perfect superelasticity between −5 °C and 15 °C. Furthermore, the effect of aging process on the superelasticity of SLM Ni-Ti alloy was systematically studied [28]. After holding at 350 °C for 1 h, the sample was predeformed at 37 °C by 6% and unloaded to produce a 4% superelastic recovery, as shown in Figure 10a. After being held at 600 °C for 1.5 h, the sample was predeformed by 6% at room temperature and unloaded to produce 5.5% superelastic recovery, as shown in Figure 10b. Gu et al. [29] believed that increasing the volume density of laser energy could reduce the coarsened Ni_4_Ti_3_ precipitates generated in situ, and significantly improved the superelastic effect of Ni-Ti alloy. Moghaddam et al. [16], for the first time, used SLM technology to form a Ni-Ti alloy capable of producing 98% superelastic recovery at 5.62% predeformation without any heat treatment, which showed excellent stability, as seen in Figure 10c,d.

In addition, many scholars have also studied the biocompatibility, damping properties and elastocaloric effect of Ni-Ti alloys formed by SLM. Karaji et al. [30] found that the effects of rhBMP2 on cell adhesion, morphology (diffusion, spindle-shaped cells), cell coverage and cell proliferation were improved in vitro by combining with porous Ni-Ti alloy formed by SLM. Cao et al. [31] obtained an excellent elastocaloric effect of 23.3 K after solution and aging treatment of Ni-Ti alloy formed by SLM, which was attributed to the large entropy change and high yield stress in the phase transformation process of Ni-Ti alloy. Zhang et al. [32] formed a Ni-Ti scaffold using SLM and fused magnesium into it to obtain Mg-NiTi composite with double continuous interpenetrating structure. The composite had high strength, high energy absorption rate and excellent damping performance at room and high temperatures.

In this section, the research status of Ni-Ti alloy formed by SLM was summarized from the aspects of Ni-Ti powder used in SLM process, SLM processing parameters, microstructure morphology, mechanical properties and functional properties of Ni-Ti alloy formed by SLM. The results showed that the microstructure of Ni-Ti alloy formed by SLM was significantly different from that of traditional processes. The grain size of Ni-Ti alloy formed by SLM was smaller, which could effectively enhance the alloy strength. The mechanical properties such as compressive strength and tensile strength reached or even exceeded the cast Ni-Ti alloy.

### 2.2. Ni-Ti Alloys Formed by LENS Technology

The thermal conductivity mode, thermal effect and molten pool morphology of LENS technology are significantly different from SLM technology. The main LENS processing parameters, such as laser power, scanning speed and powder feeding amount, have significant effects on the grain characteristics, microstructure and macroscopic properties of Ni-Ti alloy. Krishna et al. [33] used LENS technology to form Ni-Ti alloy powder with particle size of 50–150 μm, as shown in Figure 11a,b. With the increase in laser power, the grain size increased from 3.2 ± 0.8 μm to 6.3 ± 1.2 μm, and there was no obvious second phase precipitation at the grain boundary. The effect of the feeding rate on grain characteristics was further studied [34]. The results showed that, with the feeding rate increasing from 15 g/min to 30 g/min, the grain size grew significantly, as shown in Figure 11c,d. Sujith et al. [35] used LENS technology to form Ni-Ti alloy with different energy densities, and the results showed that, with the energy input gradually increasing from 27 J/mm^2^ to 70 J/mm^2^, the pore size and porosity inside the alloy gradually decreased, as shown in Figure 11e–g. Similar to SLM technology, the phase transition temperature of Ni-Ti alloy formed by LENS technology also increases with the increase in energy density, as shown in Figure 11h.

Krishna et al. [33,34] found that the compressive strength of fine grain Ni-Ti alloy formed by LENS was improved. With the increase in laser energy, the porous structure gradually changed to dense structure. The maximum recoverable strain of porous Ni-Ti alloy with 90% density was 6%, and the recoverable strain was reduced to 4% with further increase in porosity. Sujith et al. [35] found that the Ni-Ti alloy formed with energy density of 20 J/mm^2^~50 J/mm^2^ exhibited excellent superelastic effect at a low strain level (1.5%), as shown in Figure 12a. Dutkiewicz et al. [36] found that the Ni-Ti alloy formed by LENS was aged at 500 °C–2h, and the phase transition temperature A*_f_* rose after aging, making the stress platform of the aged alloy significantly decreased at room temperature. Ni-Ti alloys exhibited excellent superelastic properties at ambient temperatures higher than A*_f_* point, as shown in Figure 12b.

### 2.3. Ni-Ti Alloys Formed by Other AM Technologies

In addition to the above SLM and LENS technologies for forming Ni-Ti alloy, the technologies of additive manufactured Ni-Ti alloy also include WAAM, selective laser sintering (SLS), electron beam melting (EBM), directional energy dedication (DED) and so on. The microstructure and macroscopic properties of Ni-Ti alloy formed by different AM processes are different due to the different forming processes.

As shown in Figure 13a, Wang et al. [37] found that a large number of nano-sized Ni_4_Ti_3_ precipitates were in the Ni-Ti alloy formed by WAAM under transmission electron microscopy, and the precipitates were in the shape of lenses and distributed inside the alloy at an angle of 60° with each other. In addition, the relationship between the deposition current and the size of nano-sized Ni_4_Ti_3_ precipitates was also studied [38]. The results showed that, with the increase in deposition current from 80 A to 120 A, the average width of precipitates increased from 96.7 nm to 164.1 nm, as shown in Figure 13b. Ti_2_Ni precipitates in the Ni-Ti alloy were formed with DED [39], as shown in Figure 13c. The different microstructure had a great influence on the macroscopic properties of Ni-Ti alloy. Zeng et al. [40] evaluated the stability of Ni-Ti alloy formed by WAAM in superelastic cycle. Under the prestrain of 6%, 1.13% residual strain was generated in the first cycle, and the unrecoverable strain gradually stabilized at 2.73% as the number of cycles increased, as shown in Figure 13d,e. It was related to the inhomogeneous grain sizes and the original residual martensite in the Ni-Ti alloy. Wang et al. [39] conducted a cyclic compression test on the Ni-Ti alloy formed by DED, as shown in Figure 13f. Compared with WAAM process, the stress-strain lag area was smaller, and there was no obvious superelastic stress platform. In the first cycle, the residual strain was 0.6%, and the recoverable strain was 6.17%. With the increase in the number of cycles, the superelastic response tended to be stable, and the recoverable strain stabilized at 6.09%.

In addition, Antones et al. [41] formed Ni-Ti alloy with a diameter of 2 mm by EBM, and the results showed that the complete shape recovery rate could reach 9% when the pre-strain was between 10% and 15%. Shishkovsky et al. [42] investigated the shape memory effect of layered porous Ni-Ti alloy formed by SLS. The results showed that the porous NiTi monolayer could exhibit shape memory effect in a specific temperature range.

## 3. Application of Additive Manufactured Ni-Ti Alloys

Ni-Ti alloys have been widely used in biomedical, aerospace and other fields owing to their shape memory effect, superelastic effect, damping, biocompatibility and elastocaloric effect. For example, Ni-Ti alloy has become a kind of good biological application material because of its excellent biocompatibility, high corrosion resistance and low elastic modulus. As shown in Figure 14, a porous Ni-Ti hip prosthesis formed by SLM could be implanted into the patient’s hip bone to effectively reduce stress shielding and promote bone healing. At the same time, the best matching hip prosthesis could be customized according to the size of the patient’s bone by using the memory effect of Ni-Ti alloy [3].

Medical stents keep the anatomical structure open by supporting the inner wall so that the fluid inside the anatomical structure can flow normally. They are often used to treat diseases such as constriction of the esophagus and bile duct. As shown in Figure 15, the advantage of self-expanding porous vascular Ni-Ti alloy scaffold formed by SLM lies in its self-expanding ability, which can be curled at low temperature and limited to a small diameter and transported to the target position. After removing the restriction, the scaffold automatically unfolds and returns to the diameter of the vessel owing to the shape memory effect of Ni-Ti alloys [43]. According to the Lab 22 Additive Manufacturing Laboratory in Australia, AM technology provides freedom in stent design, enabling design developers to develop specific sizes of proximal and distal diameters of vessels, as well as larger stents, cross-branches and new shapes of proximal and distal vessels through this process. In addition, AM technology will enhance the ability of customized production of vascular stents, reduce inventory and improve the effective utilization of resources. For patients, such additive manufactured special vascular stents have better vascular compliance and are expected to improve the patient experience.

The flexible sawtooth nozzle can be adjusted by the sawtooth edge made of Ni-Ti alloy and elastic aero-alloy composite plate, as shown in Figure 16. AM technology can form a layer of Ni-Ti alloy with any thickness ratio on the titanium alloy, which can realize the design and manufacture of the serrated edge nozzle with the above complex shape [45].

## 4. Conclusions and Prospects

In this paper, the development trend of additive manufactured Ni-Ti alloy was analyzed from two aspects: the number of papers published in this field in recent years and the different AM processes for the forming of Ni-Ti alloy. The most widely used SLM process for forming Ni-Ti alloy was summarized. The relationship between materials, process, microstructure and properties of Ni-Ti alloy formed by SLM was revealed from the aspects of Ni-Ti alloy powder for SLM, SLM processing parameters, microstructure, mechanical properties and functional properties of Ni-Ti alloy. The research status of Ni-Ti alloy formed by WAAM, EBM, DED, SLS and other AM processes was briefly described, and its mechanical properties were emphatically expounded.

Based on the above research status, it can be seen that there are still gaps in the engineering application of additive manufactured Ni-Ti alloy: AM layer upon layer superposition forming process makes Ni-Ti alloy show anisotropy; the thermal stress accumulation caused by laser cyclic heating leads to many internal defects and complex microstructure characteristics of the Ni-Ti alloy during the forming process. The primary work in this field is to optimize the AM process, reveal the influence of microstructure on macroscopic properties and improve the shape memory and superelastic properties of Ni-Ti alloys. In order to accelerate the engineering application process of additive manufactured Ni-Ti alloy, further studies are suggested, as follows: design and preparation of Ni-Ti alloy powder for AM in different service environments; structure design and performance simulation analysis of additive manufactured Ni-Ti alloy; research and development of AM equipment for high precision, multi-size Ni-Ti alloy; study on the forming process of controllable changes in shape, property or functionality of additive manufactured Ni-Ti alloy; and study on the two-way, all-way shape memory effect. In particular, in terms of mechanical properties, the laws and mechanisms of anisotropy of mechanical properties of additively manufactured NiTi alloys should be clarified; the fatigue failure mechanism under superelasticity effect should be explored, and improvement methods should be proposed; and the engineering damping properties of the as-printed NiTi components should be improved from micro-regulation and macro-structure design.

## Figures and Tables

**Figure 1 materials-14-04496-f001:**
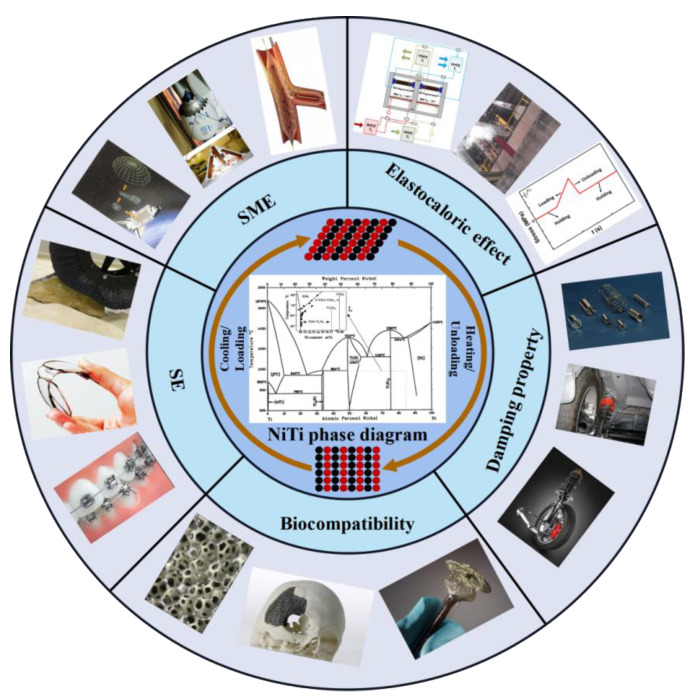
Properties and applications of Ni-Ti SMAs.

**Figure 2 materials-14-04496-f002:**
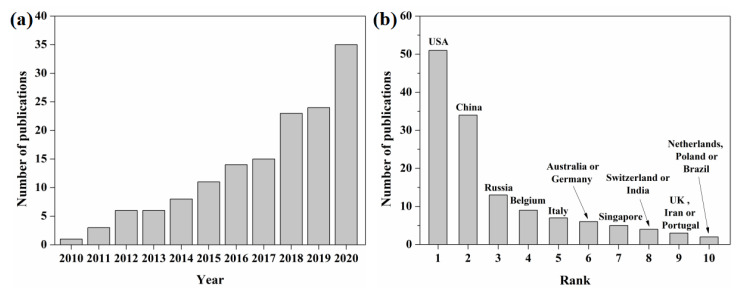
Published papers on additive manufactured Ni-Ti alloys by Web of Science: (**a**) number of papers published in different years; (**b**) number of papers published by different countries/regions.

**Figure 3 materials-14-04496-f003:**
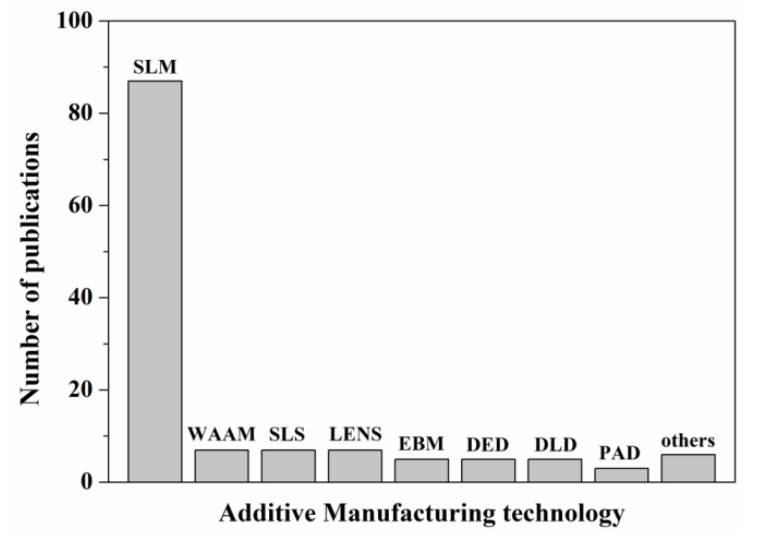
Number of papers published in the Web of Science database for the different AM technologies used to manufacture Ni-Ti alloys (SLM—Selective laser melting; WAAM—Wire and Arc Additive Manufacturing; SLS—Selective Laser Sintering; LENS—Laser Engineered Net Shaping; EBM—Electron Beam Melting; DED—Directed Energy Deposition; DLD—Direct Laser Deposition; PAD—Plasma Arc Deposition).

**Figure 4 materials-14-04496-f004:**
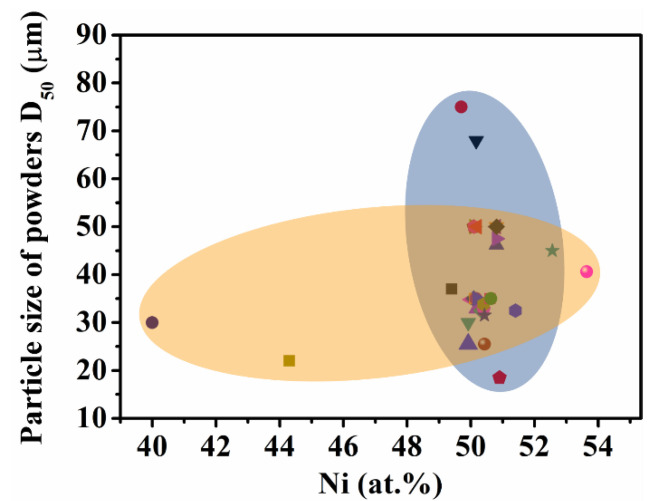
Atomic ratio and particle size of Ni-Ti alloy for SLM in the literature.

**Figure 5 materials-14-04496-f005:**
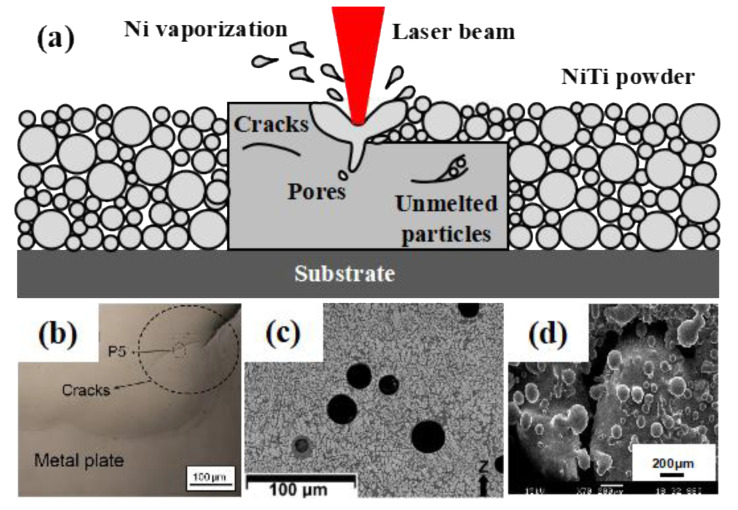
(**a**) Defects of Ni-Ti alloy formed by SLM; (**b**) the cracks; (**c**) the holes; (**d**) unfused particles [8].

**Figure 6 materials-14-04496-f006:**
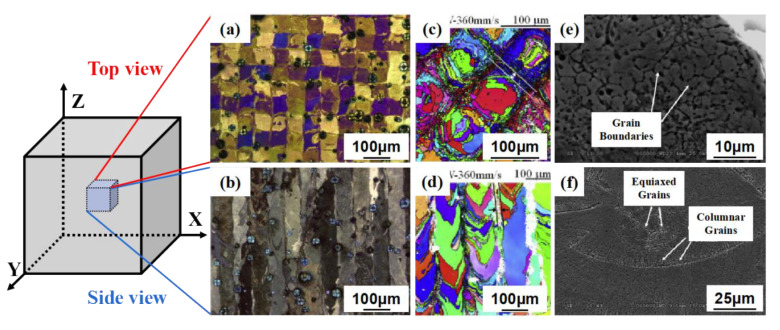
Grain morphology of Ni-Ti alloys formed by SLM parallel to and perpendicular to the depositing direction: (**a**,**b**) OM; (**c**,**d**) EBSD; (**e**,**f**) SEM [9,10,11,12].

**Figure 7 materials-14-04496-f007:**
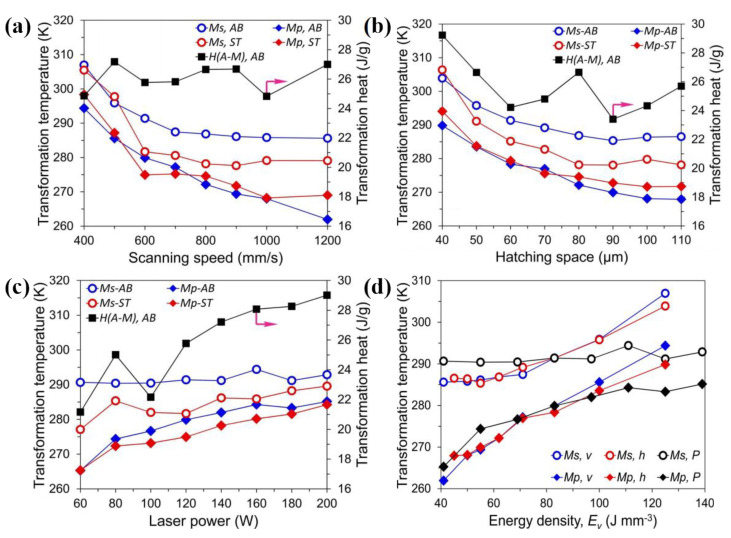
Influence of SLM processing parameters on phase transformation temperature (**a**) scanning speed; (**b**) scanning spacing; (**c**) laser power; (**d**) energy density [17].

**Figure 8 materials-14-04496-f008:**
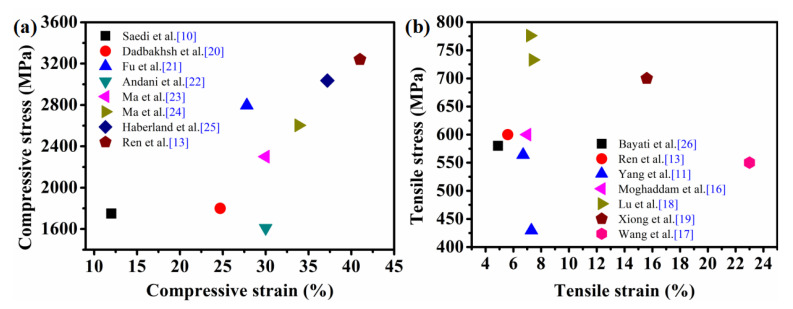
Compression and tensile properties of Ni-Ti alloy formed by SLM [10,11,13,16,17,18,19,20,21,22,23,24,25,26].

**Figure 9 materials-14-04496-f009:**
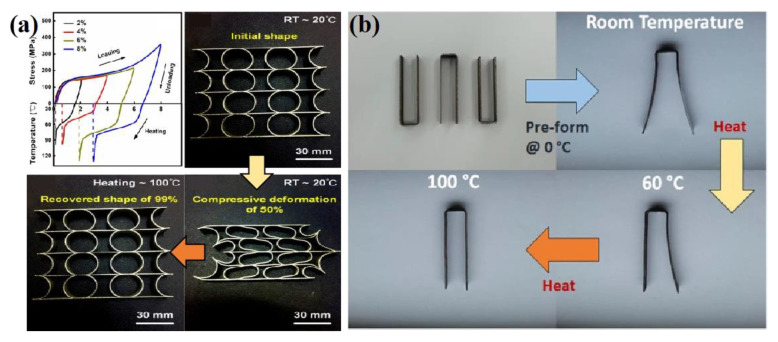
(**a**) Multilevel shape memory recovery behavior and (**b**) shape memory recovery process of Ni-Ti components formed by SLM [19,27].

**Figure 10 materials-14-04496-f010:**
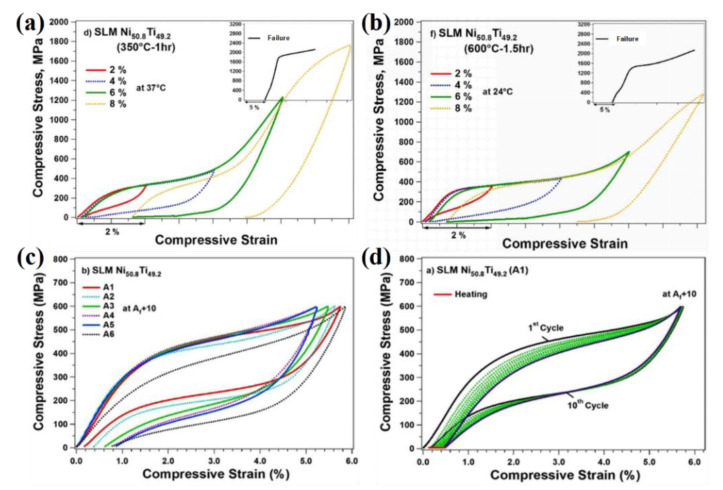
(**a**) Stress-strain curve of Ni-Ti alloys at 350 °C−1 h; (**b**) stress-strain curve of Ni-Ti alloys at 600 °C−1.5 h; (**c**) stress-strain curve and (**d**) cyclic stress-strain curve of Ni-Ti alloys without heat treatment [16,28].

**Figure 11 materials-14-04496-f011:**
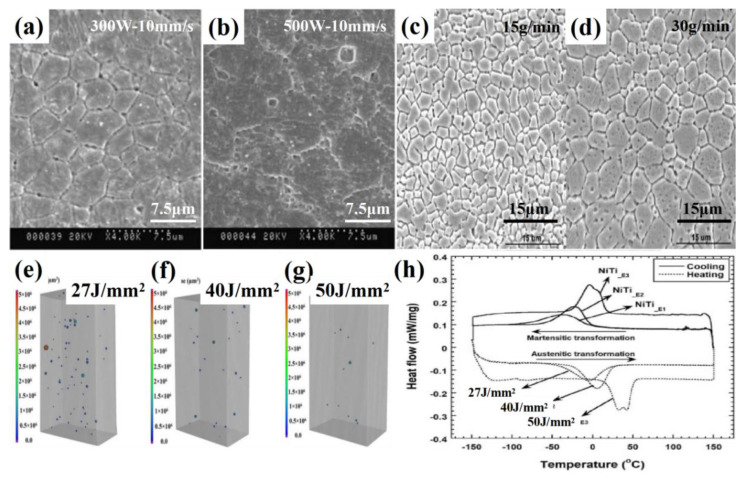
The grain morphology of Ni-Ti alloy formed by LENS under different laser powers: (**a**) 300 W–10 mm/s; (**b**) 500 W–10 mm/s; with different powder feeding rates: (**c**) 15 g/min; (**d**) 30 g/min; (**e**–**g**) porosity, pore size and (**h**) phase transition temperature of Ni-Ti alloy formed by LENS at different energy densities [33,34,35].

**Figure 12 materials-14-04496-f012:**
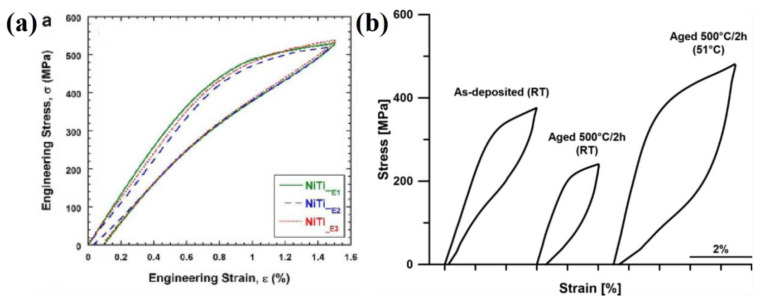
Ni-Ti alloy formed by LENS (**a**) superelastic response under different laser energy densities; (**b**) superelastic effect after aging treatment [35,36].

**Figure 13 materials-14-04496-f013:**
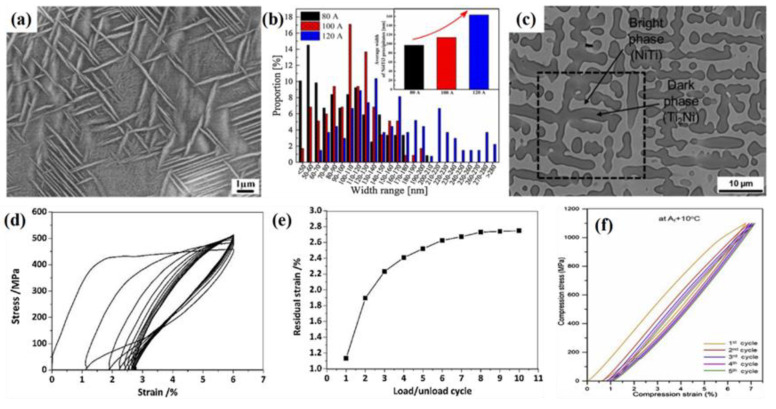
(**a**) Nano-Ni_4_Ti_3_ precipitates in the Ni-Ti alloy formed by WAAM; (**b**) the size of precipitated Ni_4_Ti_3_ phase increased with the increase in deposition current; (**c**) Ti_2_Ni precipitates in the Ni-Ti alloy formed by DED; superelastic properties of Ni-Ti alloy formed by (**d**,**e**) WAAM and (**f**) DED [37,38,39,40].

**Figure 14 materials-14-04496-f014:**
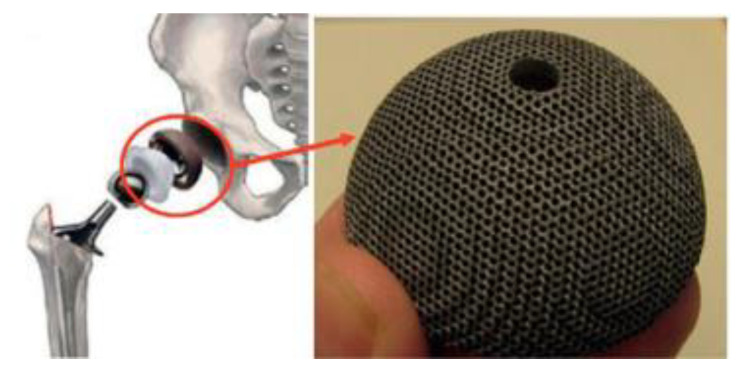
Application of porous Ni-Ti components in bone implants [3].

**Figure 15 materials-14-04496-f015:**
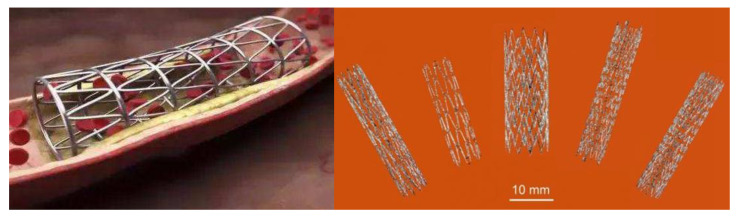
Application of Ni-Ti alloy vascular stent formed by SLM in vivo [44].

**Figure 16 materials-14-04496-f016:**
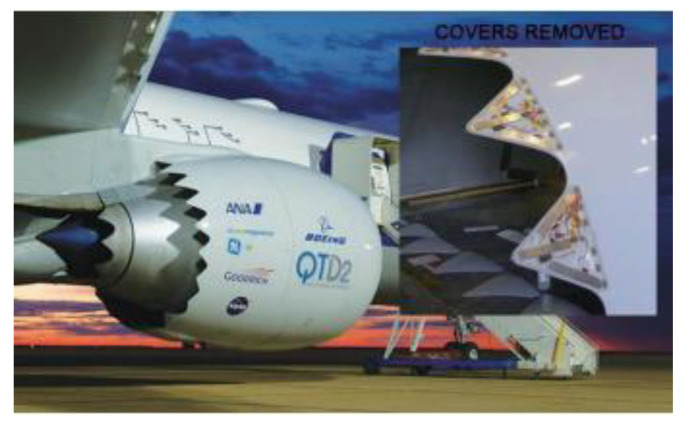
Application of Ni-Ti alloy formed by AM in aerospace field [45].

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
