# Peer review of "Research Status and Prospect of Additive Manufactured Nickel-Titanium Shape Memory Alloys"

_materials, 2021, doi:10.3390/ma14164496_

Round 1

Reviewer 1 Report

This manuscript entitled “Research status and prospect of additive manufactured nickel-titanium shape memory alloys” gives an overview. The number of papers of additively manufactured nickel-titanium (Ni-Ti) shape memory alloys (SMAs) are still not so much, but increasing year by year. In addition, additively manufactured Ni-Ti SMAs are expected for many applications. Thus, I think this manuscript is meaningful.

  1. The all abbreviations shown in Fig. 3 should be explained. Especially, the abbreviations, which are not listed in ISO 52900, need concise explanation and references.

  1. The scale bars shown in Fig. 5 (d), Fig. 6 (e), and Fig. 6 (f) cannot be recognized too small.

  1. The word of “superelastic energy” shown in the lines of 232 and 239 should be modified to better appropriate words.

Author Response

Dear Reviewer,

Thank you for your kind comments concerning our manuscript, these comments are all valuable and very helpful for revising and improving our paper, as well as the important guiding significance to our researches. We have studied comments carefully and have made correction which we hope meet with approval.

For question 1: The all abbreviations shown in Fig. 3 should be explained. Especially, the abbreviations, which are not listed in ISO 52900, need concise explanation and references.

Answer: We have explained all the abbreviations inside the figure notes in Figure 3.

For question 2: The scale bars shown in Fig. 5 (d), Fig. 6 (e), and Fig. 6 (f) cannot be recognized too small.

Answer: We have updated the scale bars in Figure 5(d) and Figure 6(e,f) to make them clearer.

For question 3: The word of “superelastic energy” shown in the lines of 232 and 239 should be modified to better appropriate words.

Answer: Thanks for the reviewer's quite good suggestion, we have carefully checked the manuscript and replaced "superelastic energy" with "superelastic effect".

Reviewer 2 Report

This is a good review and should be published. I just have minor comments on the paper format:

  1. Make sure all acronyms are defined.
  2. Polish English.
  3. Fonts in figures are hard to read, some images within figures are difficult to recognise.
  4. The connection between microstructure and properties is poor, but the topic is too broad. 

Overall this is a good paper review with a strong application and mechanical properties orientation.

Author Response

Dear Reviewer,

Thank you for your kind comments concerning our manuscript, these comments are all valuable and very helpful for revising and improving our paper, as well as the important guiding significance to our researches. We have studied comments carefully and have made correction which we hope meet with approval.

We have double-checked the manuscript and explained the abbreviations(In Fig.3) in the text; corrected some fonts in the illustrations that were not clear(In Fig. 5 and Fig.6); and polished some of the English expressions in the text to make them more readable.

It is mentioned that the unique processing process of additive manufacturing technology makes the microscopic morphology (grains, weaving and precipitated phases) inside NiTi alloy specimens different from those prepared by conventional methods, and thus has a significant impact on their macroscopic properties. The subject of this paper is the forming of NiTi alloy by additive manufacturing technology, mainly based on Selective Laser Melting and Laser Engineered Net Shaping technology, supplemented by other additive manufacturing technologies.

Reviewer 3 Report

The manuscript “Research status and prospect of additive manufactured nickel-titanium shape memory alloys” describes a study of selected scientific research and engineering application of Ni-Ti alloys. The manuscript provides a general overview of selected scientific problems. For a better explanation of the phenomena, please take note of the following comments:

  1. As is well known, the additive manufacturing technology affects the anisotropy of mechanical and fatigue properties. The authors present a comparison of compressive strength and tensile strength (Fig. 8) with a large spread of data. The practical use of these data requires an explanation of this phenomenon.
  2. Variability in the properties of additive manufacturing materials is often correlated with increased porosity and insufficient bonding of the material layers. The high surface roughness of these materials causes local stress concentration and directly translates into the premature failure of the object under fatigue loading. The manuscript should take into account these significant problems related to engineering application.
  3. The conclusions should provide detailed guidance on engineering applications in terms of material properties.

Author Response

Dear Reviewer,

Thank you for your kind comments concerning our manuscript, these comments are all valuable and very helpful for revising and improving our paper, as well as the important guiding significance to our researches. We have studied comments carefully and have made correction which we hope meet with approval.

Based on your comments, we have discussed the mechanical property data of Fig. 8 in terms of engineering applications and the study of fatigue properties of additively manufactured nickel-titanium alloys in lines 155 to 161 of the paper. In the conclusion section of the paper, with respect to the mechanical properties of additively manufactured nickel-titanium alloys, their fatigue properties are rarely reported in the literature, and we have suggested that more in-depth studies should be conducted on fatigue properties to make them better for engineering applications.

Round 2

Reviewer 3 Report

The revised manuscript does not take into account the comments from the review.

The added comment (lines 155 - 161) is too general and does not explain the phenomena described. Below are the main points that have not been commented on (previous review):

1. As is well known, the additive manufacturing technology affects the anisotropy of mechanical and fatigue properties. The authors present a comparison of compressive strength and tensile strength (Fig. 8) with a large spread of data. The practical use of these data requires an explanation of this phenomenon.
2. Variability in the properties of additive manufacturing materials is often correlated with increased porosity and insufficient bonding of the material layers. The high surface roughness of these materials causes local stress concentration and directly translates into the premature failure of the object under fatigue loading. The manuscript should take into account these significant problems related to engineering application.
3. The conclusions should provide detailed guidance on engineering applications in terms of material properties.

Please mark a comment in the revised manuscript for each point.

Author Response

Thank you for your kind comments concerning our manuscript, these comments are all valuable and very helpful for revising and improving our paper, as well as the important guiding significance to our researches. We have studied comments carefully and have made correction which we hope meet with approval.

Response to Reviewer's 3 Comments

Point 1: As is well known, the additive manufacturing technology affects the anisotropy of mechanical and fatigue properties. The authors present a comparison of compressive strength and tensile strength (Fig. 8) with a large spread of data. The practical use of these data requires an explanation of this phenomenon.

Response 1: The compressive and tensile data presented in Fig. 8 are intended to provide a chronological perspective on the continuous improvement of the compressive and tensile properties of NiTi alloys through the continuous optimization of the SLM forming process parameters. The unique processing method of laser additive manufacturing results in specimens with different mechanical properties in different loading directions. This is discussed additionally in lines 163 to 173 of the paper :

“In general, as many scholars have explored and understood more deeply about the interaction between laser and NiTi alloy materials, by continuously improving the forming process, the strength and ductility of the as-printed NiTi alloys can further meet the practical needs of engineering applications. However, from the statistical data, the wide distribution of compressive and tensile properties of NiTi alloys formed by SLM technology. Specifically, its compressive properties are much higher than its tensile properties, this is attributed to the unique layered fabrication method of additive manufacturing that leads to a preferred orientation of grain growth within the alloy, resulting in significant anisotropy of its macroscopic properties. Therefore, in engineering practice, the SLMed NiTi alloy should be applied with appropriate loading direction and loading method based on the full investigation of its polycrystalline orientation distribution.”

Point 2: Variability in the properties of additive manufacturing materials is often correlated with increased porosity and insufficient bonding of the material layers. The high surface roughness of these materials causes local stress concentration and directly translates into the premature failure of the object under fatigue loading. The manuscript should take into account these significant problems related to engineering application.

Response 2: The high porosity and the poor metallurgical bonding between the layers originate from unoptimized process parameters, which can be optimized to significantly reduce these defects; also the roughness of the component surface can be reduced by post-treatment means such as sandblasting. Of course, these internal and external defects induced by the additive manufacturing process will certainly have an impact on the fatigue performance of the component, as we discuss in additional lines 106 to 114 in the paper :

“These cracks, voids and unmelted particles caused by inappropriate process parameters will result in a certain degree of porosity within the NiTi alloy specimen and further affect the metallurgical bonding between the layers and also increase the surface roughness of the as-printed samples. When NiTi alloy components are subjected to the alternating loads, stress concentration is prone to generated at these internal or external defects, leading to the sprouting of fatigue cracks. Therefore, optimizing the forming process and investigating post-treatment methods (e.g. heat treatment, sandblasting, etc.) to reduce internal defects and external surface roughness are particularly important for the engineering applications of NiTi alloys.”

Point 3: The conclusions should provide detailed guidance on engineering applications in terms of material properties.

Response 3: In response to the reviewer's comments, we have provided detailed guidelines for engineering practical requirements for the study of mechanical properties of additively manufactured NiTi alloys in lines 354 to 359 of the paper:

“In particular, in terms of mechanical properties, the laws and mechanisms of anisotropy of mechanical properties of additively manufactured NiTi alloys should be clarified; the fatigue failure mechanism under superelasticity effect should be explored and improvement methods should be proposed; the engineering damping properties of the as-printed NiTi components should be improved from micro-regulation and macro-structure design.”